

# Response diversity of free-floating plants to nutrient stoichiometry and temperature: growth and resting body formation

Michael J. McCann

Department of Marine and Coastal Sciences, Rutgers, The State University of New Jersey, New Brunswick, NJ, United States

## ABSTRACT

Free-floating plants, like most groups of aquatic primary producers, can become nuisance vegetation under certain conditions. On the other hand, there is substantial optimism for the applied uses of free-floating plants, such as wastewater treatment, biofuel production, and aquaculture. Therefore, understanding the species-specific responses of floating plants to abiotic conditions will inform both management decisions and the beneficial applications of these plants. I measured the responses of three floating plant species common in the northeast United States (*Lemna minor, Spirodela polyrhiza,* and *Wolffia brasiliensis*) to nutrient stoichiometry (nitrogen and phosphorus) and temperature in the laboratory. I also used survey data to determine the pattern of species richness of floating plants in the field and its relationship with the dominance of this group. Floating plant species exhibited unique responses to nutrient stoichiometry and temperature in the laboratory, especially under low temperatures (18 °C) and low nutrient conditions (0.5 mg N L$^{-1}$, 0.083 mg P L$^{-1}$). The three species displayed an apparent tradeoff with different strategies of growth or dormancy. In the field, water bodies with three or more species of floating plants were not more frequently dominated by this group. The response diversity observed in the lab may not be associated with the dominance of this group in the field because it is masked by environmental variability, has a weak effect, or is only important during transient circumstances. Future research to develop applied uses of floating plants should examine response diversity across a greater range of species or clones and environmental conditions.

## INTRODUCTION

Free-floating plants (or simply "floating plants"), like most groups of aquatic primary producers, can become nuisance vegetation under certain conditions (*Portielje & Roijackers, 1995*; *Janse & Van Puijenbroek, 1998*; *Scheffer et al., 2003*; *Smith, 2012*). Shallow lakes and ponds, agricultural ditches, and tropical lakes can be dominated by thick mats of floating plants, altering abiotic conditions and reducing biotic diversity (*Morris & Barker, 1977*; *Janes, Eaton & Hardwick, 1996*; *Morris et al., 2003*; *Verdonschot & Verdonschot, 2013*).

Corresponding author
Michael J. McCann,
mccannmikejames@gmail.com

The dominance of this functional group is driven by nutrient enrichment (both nitrogen and phosphorus), and low levels of either of these nutrients can limit floating plant growth (*Portielje & Roijackers, 1995*; *Kufel et al., 2010*; *Smith, 2014*). In addition to eutrophication, increased temperatures due to climate change may also favor the dominance of this group over other primary producers (*Netten et al., 2011*; *Peeters et al., 2013*). On the other hand, there is substantial optimism for the applied uses of free-floating plants, such as wastewater treatment, biofuel production, ecotoxicological assessment, and aquaculture (e.g., *Greenberg, Huang & Dixon, 1992*; *Skillicorn, Spira & Journey, 1993*; *Ge et al., 2012*; *Xu et al., 2012*; *Verma & Suthar, 2014*). Therefore, understanding the species-specific responses of floating plants to nutrients and temperature will have both management implications and beneficial applications (*Ziegler et al., 2015*). For example, if floating plant species exhibit response diversity (i.e., unique response to abiotic conditions) (*Elmqvist et al., 2003*), then a more diverse assemblage of floating plants may be more resilient or dominant (*Naeem & Wright, 2003*), and thus harder to manage. Furthermore, if particular species have unique responses, than those with a desirable suite of traits may be identified for applied uses.

Although both nitrogen and phosphorus are important drivers of floating-plant dominance, the ratio of both nutrients may have important consequences, especially in multi-species contexts (*Smith, 2014*). Depending on the conditions of the growth medium, species and clones of floating plants differ in their N:P content (reviewed by *Landolt & Kandeler (1987)*). For example, *Karpati & Pomogyi (1979)* reported N:P tissue content ranging from 2.65 for *Lemna trisulca* to 10.53 for *Lemna minor* in naturally growing plants. *Docauer (1983)* reported N:P content of 8.12 for *Spirodela polyrhiza*, 10.38 for *L. minor,* and 3.46 and 6.54 for two species of *Wolffia* (*W. borealis* and *W. columbiana,* respectively) when the plants were growing at half of their maximum growth rate. Depending on the nutrient content of the growth medium, tissue N:P in *Lemna gibba* can range from approximately 3 to nearly 40 (*Fulton et al., 2010*). These species-specific and context-dependent stoichiometric differences are important because nutrient stoichiometry will differ depending on the source of nutrient loading and various other factors, resulting in wide variation in nutrient stoichiometry of different water bodies (*Downing & McCauley, 1992*). If floating plant species are constrained in their nutrient stoichiometry, then this may affect the outcome of competition among floating plants or with other primary producer groups (*Sterner & Elser, 2002*), although nitrogen alone explained most of the outcome of competition among floating plants in the early stages of a field mesocosm experiment (*Smith, 2014*).

I used laboratory experiments to determine the response diversity (i.e., unique response to abiotic conditions) among floating plant species. To address this question, I performed two laboratory experiments to examine the growth and turion-formation of floating plant species in response to nitrogen, phosphorus, and temperature. The experiments were conducted with three of the most common floating plant species in the northeast United States: *Lemna minor* L., *Spirodela polyrhiza* (L.) Schleid., and *Wolffia brasiliensis* Wedd. To understand whether the response diversity in the laboratory corresponds to increased dominance in the field, I also analyzed a dataset of over 200 freshwater lakes and ponds to

determine the association between floating plant species composition and richness and the occurrence of floating plant dominance. If substantial differences exist among floating plant species in the lab, then a more diverse assemblage may be more likely to be dominant over a broader range of conditions. In this case, it is expected that water bodies with greater floating plant richness will be more frequently found in a floating plant state.

## MATERIALS AND METHODS

### Laboratory conditions

Although conducting a single, large experiment to determine the response to all environmental conditions of interest would have some advantages, I conducted two separate experiments to allow for a manageable amount of effort and maintain adequate sample sizes in each experiment. For all experiments, plants were collected from Setauket Mill Pond, East Setauket, New York, USA (40.946061°, −73.115613°) and acclimated under experimental conditions prior to the start of the experiments. Plants were collected in mid-August or mid-June for experiments I and II, respectively. Species were identified according to *Crow & Hellquist (2000)*. Modified Barko-Smart media (*Smart & Barko, 1985*; *Szabó et al., 2005*) with phosphorus supplied as potassium dihydrogen phosphate and nitrogen supplied as a 1:1 ratio of nitrogen from nitrate and ammonium (potassium nitrate and ammonium chloride) was used as the nutrient media. Micronutrients were supplied to the media by Tropica Aquacare Plant Nutrition Liquid at a concentration of 0.1 mL L$^{-1}$ (*Szabó, Roijackers & Scheffer, 2003*; *Szabó et al., 2010*). Plants were grown in plastic, multiwell plates with individual well diameters of 22.75 mm containing 4 mL of media. Each well housed a single replicate of an experimental treatment. Multiwell plates were cleaned in 10% hydrochloric acid for at least 1 hr, and then rinsed thoroughly with deionized water prior to using. Living plants (i.e., green) were moved to clean multiwell plates with fresh media every two to four days, depending on the experiment. Any dead fronds (i.e., entirely white or brown) were removed. Light was supplied at an intensity of 130–150 $\mu$E m$^{-2}$ s$^{-1}$ and a 14:10 hr light:dark photoperiod, which is within the range of many previous studies (reviewed by *Landolt & Kandeler (1987)*). Temperature-controlled walk-in chambers were used to achieve the target temperatures. Nutrient treatments and species were systematically assigned to wells to ensure that replicates were dispersed across plates and not in adjacent wells. A systematic assignment of treatments, rather than completely randomized, was used to reduce the likelihood that the wrong nutrient medium or plant species would be placed in a well during each nutrient media change. Because of the large number of replicates and the small spatial scale of the experimental setup, it is unlikely that the systematic assignment of treatments correlated with an unknown, confounding variable that varied along the same systematic spatial pattern of treatment assignments. Initial plant area (20 mm$^2$ or 5% of the total well area available for growth) was approximately equal for all species within an experiment. Frond number differed because of the size differences among species. In order to prevent crowding, experiments were ended when plants in some treatments filled approximately two-thirds of the well area.

Raw data from both experiments are available in Supplemental Informations 1 and 2.

## Experiment I: response to nutrients and temperature

I measured the responses of floating plant species to nutrients and temperature by measuring their growth (in terms of surface area) and turion formation at three nutrient levels (low: 0.5 mg N L$^{-1}$ and 0.083 mg P L$^{-1}$; medium: 5 mg N L$^{-1}$ and 0.83 mg P L$^{-1}$; or high: 10 mg N L$^{-1}$ and 1.66 mg P L$^{-1}$) fully-crossed with three temperatures (18, 24, and 30 °C), for a total of nine treatment combinations. The three nutrient treatments are all at a N:P mass ratio of ~6 and correspond to experimental treatments used in previous studies (e.g., *Scheffer et al., 2003*; *Szabó et al., 2010*). At the lowest nutrient level, nutrients were expected to be limiting to growth (*Lüönd, 1983*; *Szabó et al., 2010*), but for the two highest nutrient levels, nutrients may be saturated (*Szabó et al., 2010*). Although this is not an exhaustive combination of treatments, these levels sample some of the possible environmental conditions encountered by floating plants in nature, and potentially in engineered applications (e.g., wastewater treatment). Eight replicates of each treatment combination for each species were grown for 12 days. Plants were transferred to new nutrient media 3, 5, 7, and 10 days after the start of the experiment.

## Experiment II: response to nutrient stoichiometry

In a second experiment, I measured the responses of floating plant species to nutrient stoichiometry by measuring their growth and turion formation at all nine combinations of three nitrogen (0.5, 5, and 10 mg N L$^{-1}$) and three phosphorus levels (0.083, 0.83, and 1.66 mg P L$^{-1}$), producing a variety of N:P mass ratios, ranging from 0.30 to 120.48 (Table 1). At the lowest treatment level of each nutrient, that nutrient may limit plant growth. Six replicates of each plant species at each of the nine nutrient treatments were grown for 17 days. A smaller number of replicates per each treatment combination (6 vs. 8 replicates) were used in this experiment because of the relatively low variance observed in Experiment I. I transferred plants to new nutrient media 3, 7, 10, and 14 days after the start of the experiment. Plants were grown at approximately 30 °C, which had similar growth rates as the 24 °C treatment and resulted in the maximum growth rates in Experiment I. This temperature would ensure that only nutrient stoichiometry would limit plant growth in this experiment.

## Growth and turion production

To quantify growth, plants in each replicate of each treatment were photographed (Nikon Coolpix 5700 Digital Camera) with backlighting from a light box (Laboratory Supply Company, 60 Watts) on days that nutrient media were changed. The two-dimensional plant area on the water surface was measured with ImageJ, version 1.47 (*Rasband, 1974–2014*), using the threshold function on an 8-bit grayscale photo, after dead fronds had been removed by hand (see above). Relative Growth Rate (RGR) was calculated between each measurement. RGR was calculated as $[\ln(A_2) - \ln(A_1)]/(t_2 - t_1)$, where $A$ is the area of plants in mm$^2$, $t$ is time in days, and subscripts 2 and 1 indicate two sequential time points in the experiment. Plant thickness or mass was not measured during these experiments, but the overheard surface area is a commonly used measure in many experiments (reviewed by *Landolt & Kandeler (1987)*). Turions were distinguished

**Table 1 N:P mass ratios produced by nine combinations of nitrogen and phosphorus at 30 °C in Experiment II.**

| | | Nitrogen (mg L$^{-1}$) | | |
| --- | --- | --- | --- | --- |
| | | 0.5 | 5 | 10 |
| Phosphorus (mg L$^{-1}$) | 0.083 | 6.02 | 60.24 | 120.48 |
| | 0.83 | 0.60 | 6.02 | 12.05 |
| | 1.66 | 0.30 | 3.01 | 6.02 |

as plants that had sunk to the bottom of the experimental vessel, and they typically differed in size, texture, or color from plants on the surface. The number of turions (i.e., asexual resting bodies) produced were counted for each experimental replicate after the live plants had been moved to fresh media. The number of turions was converted to area with equations developed in another study (Supplemental Information 3). In some replicates, all plants in a replicate died (i.e., bleached white) during the experiment, and were re-started with new plants, assuming that the failed growth was due to damage to the plant when handling. These replicates were excluded from analysis if they were not grown for at least 10 days.

## Statistical analysis

For all ANOVAs, data were tested for normally distributed residuals with a Shapiro-Wilk test and equal variance among treatment groups with Levene's test. If the data did not meet these assumptions of ANOVA, they were power transformed to ensure that these criteria were met. I performed all statistical analyses in R version 3.0.2 (*R Development Core Team, 2013*).

   The goal of these experiments was to test whether floating plant species differed in their response to environmental conditions and under what conditions they differed. Therefore, analyses tested for differences between species under particular combinations of environmental conditions (i.e., treatment levels). For both experiments, I performed a one-way ANOVA for each treatment combination to test for an effect of species on the average RGR. I used a Dunn-Šidák correction to adjust p-values for multiple comparisons. When significant treatment effects were found, Tukey's HSD was used to detect differences among species. An alternative approach to analyzing these data is to use factorial ANOVAs to test for the main and interactive effects of species and experimental conditions on growth rates (Supplemental Information 4). Since the number of possible pairwise, posthoc comparisons in each experiment is large (351) and species differences under identical conditions (i.e., response diversity) was the focus of this study, this statistical approach is not reported in the main text.

   In the first experiment (the effect of nutrients and temperature) only *W. brasiliensis* formed turions. I analyzed the effect of nutrients and temperature on turion area produced (mm$^2$ day$^{-1}$) by *W. brasiliensis* with a two-way ANOVA. When significant treatment effects were found, I used Tukey's HSD to detect differences among treatment levels. In the second experiment (i.e., the effect of nitrogen and phosphorus),

*W. brasiliensis* formed turions under all treatment levels and *S. polyrhiza* under some treatment levels. To detect differences in turion production rate between species at particular nutrient levels, I used one-way ANOVAs at each nutrient level where both *W. brasiliensis* and *S. polyrhiza* formed turions. I used a Dunn-Šidák correction to adjust p-values for multiple comparisons.

### Floating plant richness and abundance in natural water bodies

I examined the occurrence and dominance of floating plant species in lakes and ponds with a dataset of 205 freshwater water bodies in Connecticut and Long Island, NY (Supplemental Information 5). The data came from two sources: 1) 184 surveys by the Connecticut Agricultural Experiment Station (CAES) in 2005 to 2013, and 2) 21 surveys that I conducted in Long Island, New York and Connecticut, USA in 2011 to 2013. This data set spanned a range of perennial, freshwater lakes and ponds and included the list of floating plant species present in the water body and the maximum floating plant cover (percent of water body covered as quantified through visual observation and mapping) during the late summer (late July to September). See *Capers et al. (2007)* for a description of the survey methods used for the CAES data. For the Long Island surveys, plant cover was estimated through visual observation similar to methods used in previous studies (*Driever, Van Nes & Roijackers, 2005*; *Smith, 2012*). In this study, I use high floating plant cover as a surrogate for dominance by floating plants, while acknowledging that a consideration of other primary producers (e.g., phytoplankton, submerged vegetation) and covariates is necessary for a rigorous demonstration of complete floating-plant dominance.

I used a goodness of fit test (G-test, based on a chi-square) to test if all floating plant species richness levels were equally likely to occur (i.e., random), excluding water bodies without floating plants. I used a second G-test to determine if floating plant dominance ($\geq$66.67% cover) was equally likely to occur under different levels of floating plant richness. The expected value for each richness level in water bodies dominated by floating plants was based on the observed frequency of each floating plant richness level across all water bodies (both dominated and non-dominated). Floating plant richness was categorized as 1, 2, or $\geq$3 species to ensure adequate sample sizes in each level.

## RESULTS

### Experiment I: response to nutrients and temperature

Average relative growth rate (RGR) was different among species at six of the nine combinations of nutrients and temperature (Table 2; Fig. 1). Species growth rates were equal when nutrients and temperatures were high (10 mg N L$^{-1}$ and 1.66 mg P L$^{-1}$ and 24 or 30 °C) or at 18 °C and medium nutrients (5 mg N L$^{-1}$ and 0.8 mg P L$^{-1}$). Typically, *Lemna minor* and *Spirodela polyrhiza* growth rates were equal to each other and both were greater than the growth rate of *Wolffia brasiliensis* (Fig. 1). Only *W. brasiliensis* formed turions in this experiment. There was a significant effect of nutrients ($F_{2,64} = 4.770$, $p = 0.012$), temperature ($F_{2,64} = 38.706$, $p < 0.001$), and significant interaction ($F_{4,64} = 4.089$, $p = 0.005$) on the turion production rate of *W. brasiliensis* (Fig. 2). At both

**Table 2** One-way ANOVAs for the effect of species on the average relative growth rate (RGR) of floating plants at nine combinations of nutrients and temperature.

| Treatment | | Average RGR | |
|---|---|---|---|
| Nutrients | Temperature (°C) | F-statistic | p-value |
| Low | 18 | 11.403 | <0.001 |
| | 24 | 39.83 | <0.001 |
| | 30 | 30.14 | <0.001 |
| Medium | 18 | 5.703 | 0.011 |
| | 24 | 7.172 | 0.004 |
| | 30 | 8.136 | 0.002 |
| High | 18 | 12.44 | <0.001 |
| | 24 | 4.106 | 0.031 |
| | 30 | 4.325 | 0.027 |

Note:
Degrees of freedom for all ANOVAs were 2 and 21, except for at low nutrients and 30 °C, where df = 2, 20. Dunn-Šidák adjusted critical p-value is 0.0057. Nutrient levels are low = 0.5 mg N $L^{-1}$ and 0.083 mg P $L^{-1}$, medium = 5 mg N $L^{-1}$ and 0.83 mg P $L^{-1}$, or high = 10 mg N $L^{-1}$ and 1.66 mg P $L^{-1}$.

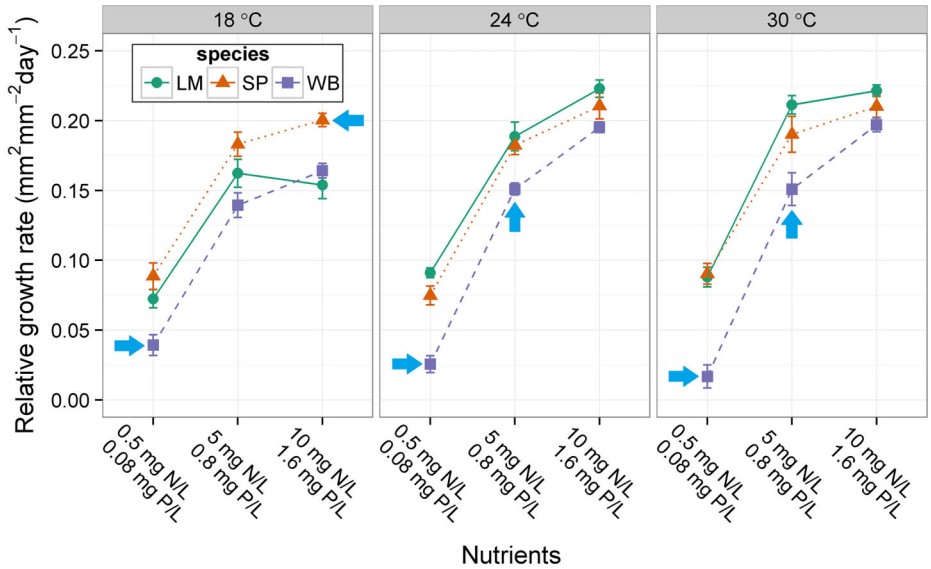

**Figure 1 Effect of nutrients and temperature on relative growth rate (RGR) of three species of floating plants.** Error bars are standard errors. Post-hoc comparisons among species are for each response variable at each level of nutrients and temperature. Arrows indicate a species that is statistically different (Tukey's HSD, p > 0.05) at a given nutrient and temperature level. LM, *Lemna minor*; SP, *Spirodela polyrhiza*; WB, *Wolffia brasiliensis*.

18 and 30 °C *W. brasiliensis* decreased turion production at the highest nutrient level, but at 24 °C turion production increased with nutrient level (Fig. 2).

## Experiment II: response to nutrient stoichiometry

Average RGR differed among species in four of the nine nutrient combinations (Table 3; Fig. 3). Species differences were found whenever nitrogen was low (0.5 mg N $L^{-1}$) or when

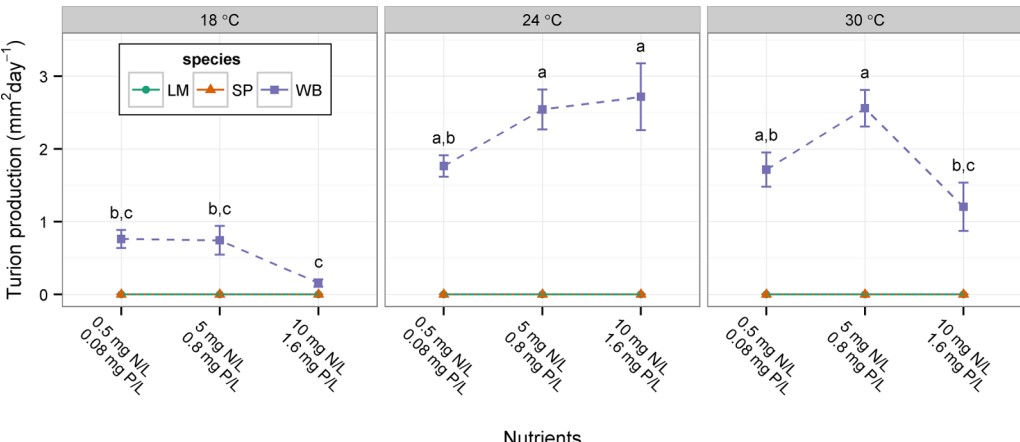

**Figure 2 Effect of nutrients and temperature on resting body formation.** Error bars are standard errors. Shared letters indicate no difference between treatment levels (Tukey's HSD, p > 0.05) for *W. brasiliensis* turion production. LM, *Lemna minor*; SP, *Spirodela polyrhiza*; WB, *Wolffia brasiliensis*.

**Table 3 One-way ANOVAs for the effect of species on the average relative growth rate (RGR) of floating plants at nine combinations of nitrogen and phosphorus at 30 °C.**

| Treatment | | Average RGR | |
|---|---|---|---|
| Nitrogen | Phosphorus | F-statistic | p-value |
| Low | Low | 21.24 | <0.001 |
| | Medium | 60.61 | <0.001 |
| | High | 14.1 | <0.001 |
| Medium | Low | 14.08 | <0.001 |
| | Medium | 0.985 | 0.396 |
| | High | 1.666 | 0.222 |
| High | Low | 0.506 | 0.613 |
| | Medium | 4.727 | 0.026 |
| | High | 1.283 | 0.306 |

**Note:**
Degrees of freedom for all ANOVAs were 2 and 15. Dunn-Šidák adjusted critical p-value is 0.005. Nitrogen levels are low = 0.5 mg N L$^{-1}$, medium = 5 mg N L$^{-1}$, and high = 10 mg N L$^{-1}$. Phosphorus levels are low = 0.083 mg P L$^{-1}$, medium = 0.83 mg P L$^{-1}$, and high = 1.66 mg P L$^{-1}$.

phosphorus was low and nitrogen was medium (0.08 mg P L$^{-1}$ and 5 mg N L$^{-1}$). Both *S. polyrhiza* and *W. brasiliensis* formed turions in this experiment. *W. brasiliensis* formed turions at all combinations of nitrogen and phosphorus, whereas *S. polyrhiza* only formed turions at low nitrogen and medium and high phosphorus or low phosphorus and medium and high nitrogen (Fig. 4). At low nitrogen and medium phosphorus (ANOVA, $F_{1,9}$ = 51.62, p < 0.001), low nitrogen and high phosphorus (ANOVA, $F_{1,10}$ = 49.91, p < 0.001), and medium nitrogen and low phosphorus (ANOVA, $F_{1,10}$ = 49.82, p < 0.001), *W. brasiliensis* had a greater turion production rate than *S. polyrhiza* (Fig. 4). Only at low phosphorus and high nitrogen both species had equal turion production rates (ANOVA, $F_{1,6}$ = 6.64, p = 0.042) (Dunn-Šidák adjusted critical p-value 0.013).

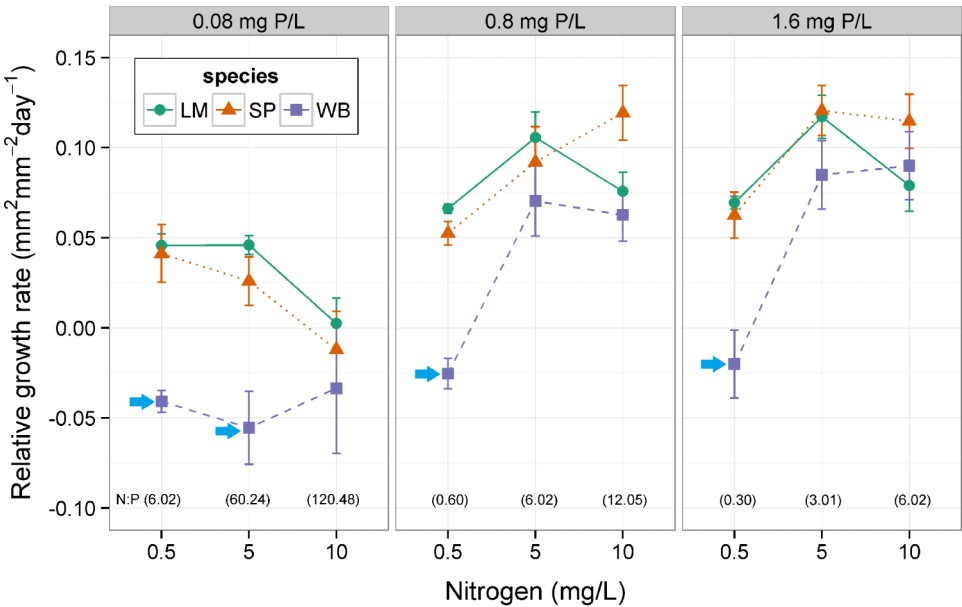

**Figure 3 Effect of nitrogen and phosphorus on relative growth rate (RGR) of three species of floating plants at 30 °C.** Error bars are standard errors. Post-hoc comparisons among species are for each response variable at each level of nitrogen and phosphorus. Arrows indicate a species that is statistically different (Tukey's HSD, p > 0.05) at a given nutrient and temperature level. N:P ratios are indicated in parentheses above the horizontal axis. LM, *Lemna minor*; SP, *Spirodela polyrhiza*; WB, *Wolffia brasiliensis*.

## Floating plant richness and abundance in natural water bodies

Most freshwater lakes and ponds in Connecticut and Long Island, NY did not have any floating plants present (106 of 205, Table 4). Across all water bodies with floating plants present, a total of seven taxa were found. *L. minor*, *S. polyrhiza*, and *Wolffia* spp. were the most common taxa, occurring in 82, 47, and 42 of the 205 water bodies, respectively. The next most common species, *L. trisulca*, only occurred in 4 lakes and ponds. Among water bodies with floating plants present, the occurrence of different levels of species richness levels was non-random (Table 4; Fig. 5A, G-test, G = 6.909, df = 2, p < 0.031). Monocultures were more common than expected and three- and four-species polycultures were less common than expected (Table 4; Fig. 5A).

Only twenty water bodies had floating plant cover greater than 66.67%. Among these water bodies, there was no significant association between floating plant richness categories and the frequency of occurrence (Table 4; Fig. 5B, G-test, G = 2.430, df = 2, p = 0.7). Although not statistically significant, water bodies dominated by floating plants tended to have three or more species of floating plants, whereas water bodies not dominated by floating plants tended to have one or two species (Fig. 5B). The results of these analyses did not change if a higher threshold for floating plant dominance (e.g., 80% cover) was used or if the analysis was limited to small water bodies (<5 ha surface area) or water bodies with higher nutrients (total phosphorus >0.02 mg P L$^{-1}$).

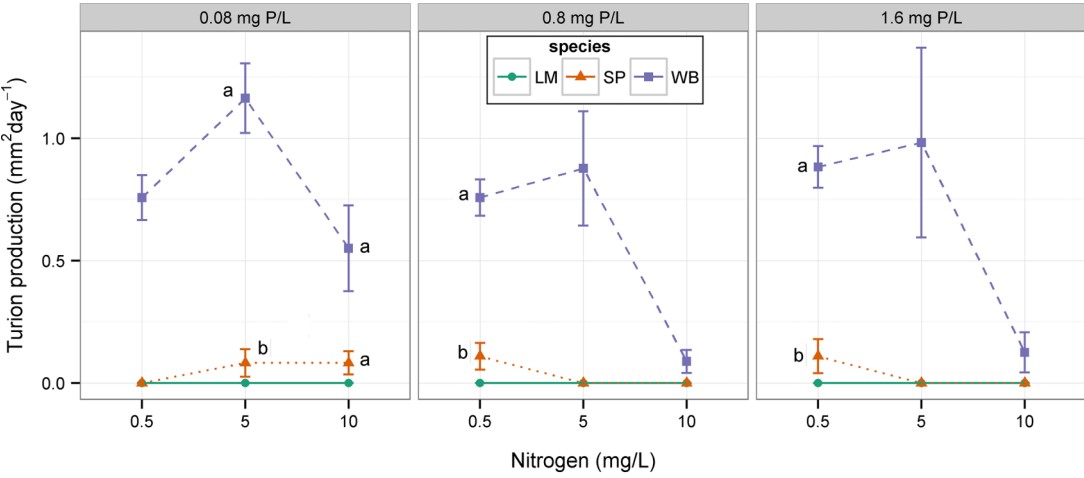

**Figure 4 Effect of nitrogen and phosphorus on turion formation of three species of floating plants.** When more both *L. minor* and *S. polyrhiza* produced turions at a particular treatment level, significant differences between those species are indicated by unique letters (Tukey's HSD, p > 0.05). Error bars are standard errors. LM, *Lemna minor*; SP, *Spirodela polyrhiza*; WB, *Wolffia brasiliensis*.

**Table 4 The frequency of floating plant species compositions and the frequency of floating plant cover exceeding two-thirds of the surface area of freshwater lakes and ponds in Connecticut and Long Island, NY.**

| Floating plant species richness | Species composition | Frequency of occurrence | Frequency floating plant cover >66.67% |
|---|---|---|---|
| 4 | A, **LM, SP, W** | 1 | 1 |
| | **LM**, LV, **SP, W** | 1 | 0 |
| | **LM**, R, **SP, W** | 1 | 0 |
| 3 | **LM**, LT, **SP** | 2 | 0 |
| | **LM, SP, W** | 18 | 7 |
| | LT, **SP, W** | 1 | 0 |
| | All ≥3 species polycultures | 24 | 8 |
| 2 | A, **W** | 1 | 1 |
| | **LM, SP** | 14 | 2 |
| | **LM, W** | 13 | 2 |
| | **SP, W** | 2 | 0 |
| | All 2 species polycultures | 30 | 5 |
| 1 | A | 1 | 1 |
| | **LM** | 32 | 4 |
| | LT | 1 | 0 |
| | **SP** | 7 | 0 |
| | **W** | 4 | 2 |
| | All monocultures | 45 | 7 |
| 0 | None | 106 | 0 |
| | TOTAL | 205 | 20 |

Note:
A, *Azolla* sp.; **LM**, *Lemna minor*; LT, *L. trisulca*; LV, *L. valdiviana*; R, *Riccia* sp.; **SP**, *Spirodela polyrhiza*; **W**, *Wolffia* sp. Taxa used in the laboratory experiments are indicated by bold letters.

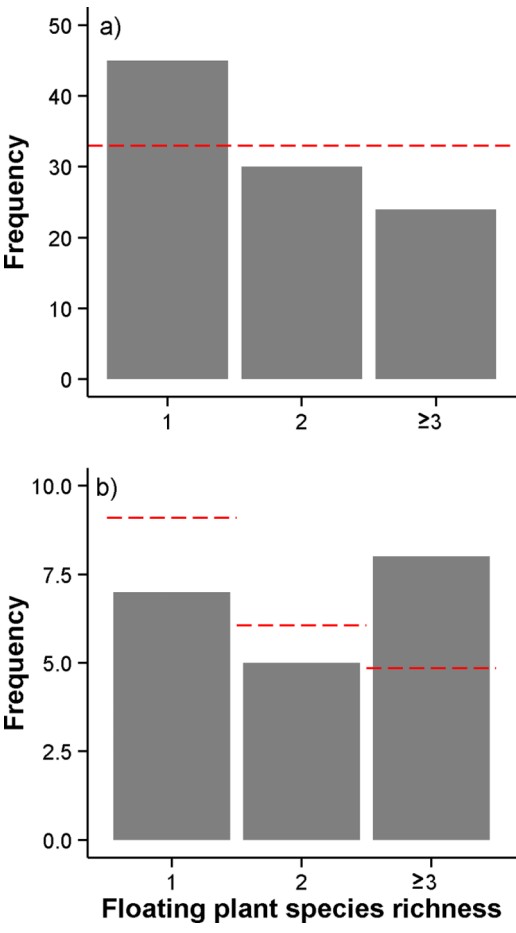

**Figure 5 Floating plant species richness in lakes and ponds in Connecticut and Long Island, NY, USA.** (A) All water bodies with floating plants present (n = 99), and (B) water bodies with floating plant cover >66.67% of the water surface (n = 20). Dashed lines indicate expected value if random.

## DISCUSSION

In general, the floating plant species in this study exhibited differences (i.e., response diversity) in their average growth rates across nitrogen, phosphorus, and temperature conditions. Differences among species were typically seen under less favorable conditions (i.e., low nutrients and low temperatures), whereas species typically had similar growth rates under conditions expected to be most favorable for their growth (i.e., high nutrients and high temperature). When differences were detected, *Lemna minor* and *Spirodela polyrhiza* typically grew at rates that were equal to each other and higher than *Wolffia brasiliensis*. On the other hand, *W. brasiliensis* produced more resting bodies across most experimental conditions, whereas *S. polyrhiza* only occasionally produced turions. *L. minor* never produced turions in these experiments. This suggests a tradeoff between producing floating or sinking biomass under these experimental conditions and may explain the lower relative growth rate of *W. brasiliensis*. In the field, a floating plant-dominated state did not occur more frequently in water bodies with higher floating plant
richness, opposite to the expectation if response diversity is important for formation of floating plant dominance.

The apparent tradeoff between growth and resting body production among floating plant species may have important consequences for this functional group. The different strategies among species can allow the floating plant functional group as a whole to have both rapid growth on the water surface and insurance against perturbation via their resting bodies. Therefore, floating plant polycultures may have a combination of strategies that may not be achievable by a single species. For example, the floating plant functional group in a water body with both *L. minor* and *W. brasiliensis* could have both faster growth at low nutrients or temperatures (due to the traits of *L. minor*) and a greater number of resting bodies to re-colonize the water body at the start of a growing season (due to the traits of *W. brasiliensis*). A polyculture of floating plant species and their unique responses to environmental conditions may allow the functional group to attain higher biomass or persist in a water body over a broader range of conditions.

Although response diversity in the lab did not translate to a correlation between floating plant-dominance and higher floating plant richness in the field, the response diversity may be relevant for applied uses of these plants (e.g., wastewater treatment, biofuel production, and aquaculture). While the optimal species or combination of species will depend on the particular application, some general patterns exist. For example, if maximization of floating plant surface growth is the sole objective (e.g., a highly controlled application where biomass is harvested), then either *L. minor* or *S. polyrhiza* would be a better choice than *W. brasiliensis* because of their higher growth rates. Future experiments would need to determine whether a combination of these two species or a monoculture would be optimal. On the other hand, if formation of resting bodies was a desirable property in the application (e.g., perennial, outdoor uses), then including *S. polyrhiza, W. brasiliensis,* or another species that produces turions would be recommended. For some applied uses of floating plants, other types of experiments need to be conducted. For example, use in combined aquaculture systems with fish would require an understanding of the preference and nutritional attributes of plant species with regards to the fish species.

The lack of a relationship between response diversity of this functional group and its ability to become dominant may be due to a variety of factors. In the field, environmental variability may outweigh the relationship between species richness and the formation of the floating plant state. In addition to response diversity, other factors will influence the occurrence of the floating plant state in the field. For example, water bodies in the northeast United States above a size threshold (~5 ha) are rarely dominated by floating plants (McCann, 2014, personal observations). It is also possible that the floating plant response diversity observed in the laboratory only has a small effect on dominance in the field. Although species differences are quantifiable in the laboratory, their magnitude may not be large enough to be detectable in the field. Furthermore, since floating plant dominance is relative uncommon in this region (<10% of water bodies), there may be low statistical power to detect a relationship between species richness and floating plant dominance, especially if effect sizes are small.

It is also possible that response diversity is only important during transient circumstances or conditions rarely encountered in these surveys. Therefore, the response diversity exhibited by floating plant species in the lab will not determine whether this functional group is currently in a dominant state in the field. As a result, species-rich water bodies could lose floating plant species due to local extinction and still maintain floating plant dominance. Response diversity may also help this functional group form a dominant state in other geographic regions where low temperatures and low nutrients are more common (e.g., the Upper Midwest United States). Rather than allowing floating plants to achieve dominance, the response diversity observed may help this functional group persist in a water body, despite unfavorable conditions. Interestingly, *L. minor,* which did not produce resting bodies in the lab but typically had the fastest growth rate (along with *S. polyrhiza*), was the most common floating plant species in this region (present in 82 of 205 water bodies). Despite differences in growth rates and resting body production, *S. polyrhiza* and *W. brasiliensis* occurred in a similar number of water bodies (47 and 42, respectively).

The lower growth rates observed in Experiment II (Fig. 3) relative to Experiment I (Fig. 1) may be due to the fact that the nutrient media was changed less frequently (every 3–4 days compared to 2–3 days) and nutrient levels likely decreased to a greater extent between media changes in Experiment II. Also, *S. polyrhiza* only produced turions in Experiment II. The difference in media change frequency may have caused the difference in turion production between experiments, or there may be differences based on the timing when plants were collected from the field (Mid-August and mid-June for Experiments I and II, respectively). Therefore, strict comparison of the growth rates or turion production between experiments should not be done without consideration of the differences in experimental conditions.

There are few others studies of the response diversity of the floating plant functional group to temperature, nutrients, or other environmental variables. *Lüönd (1983)* measured the response of *L. minor, S. polyrhiza,* and two other species of *Lemna* to nitrogen and phosphorus at 25 °C. All species increased their growth rates in response to increases of both nutrients, as in this study, and all species decreased their growth rate at extremely high nutrient levels (e.g., 1.75 g N L$^{-1}$, 1.36 g P L$^{-1}$) (*Lüönd, 1983*). The presence, but not the rate, of turion production was reported for *S. polyrhiza.* Unfortunately, no statistical comparisons were made to determine if species had unique responses under particular conditions (*Lüönd, 1983*). *Lemon, Posluszny & Husband (2001)* examined the growth of *L. minor, S. polyrhiza,* and *W. borealis,* a cogener of *W. brasiliensis* at 24 °C and very high nutrients (33% v/v Hutner's medium, ~31 mg N L$^{-1}$, ~23 mg P L$^{-1}$), and found that *W. borealis* has the highest growth rate, while *S. polyrhiza* has the lowest (in terms of frond number, not area growth rate). Results of turion production were not reported (*Lemon, Posluszny & Husband, 2001*). Some studies have examined response diversity to variables not included in this study. Floating plants appear to have response diversity to pH (*Hicks, 1932*; *McClay, 1976*). *Lemna minor, Spirodela oligorrhiza,* and *Wolffia arrhiza* all have a similar pH range (pH ~3 to 10), but their optimal pH differs, from mildly acidic (*W. arrihiza,* pH 5.0 or *L. minor* pH 6.2) to neutral (*S. oligorrhiza,*

pH 7.0) when grown in the lab at 25 °C at very high nutrients (~241 mg N L$^{-1}$ and ~32 mg P L$^{-1}$) (*McClay, 1976*).

While this study was only able to examine a subset of all floating plant species under particular combinations of environmental conditions, it found some conditions where species have response diversity and others where species are redundant. Further studies, including a greater number of species or clones and environmental variables, as well as determining tradeoffs between responses (e.g., growth or resting bodies), are necessary to determine the full breadth of response diversity of this functional group. For example, previous work on floating plant performance under low temperature conditions (~10 °C) shows that species differ in their minimum temperature (*Landolt & Kandeler, 1987*), which may have important consequences for growth of this functional group at the beginning and end of a growing season. While *Ziegler et al. (2015)* examined the maximum relative growth rate of 39 clones of 13 species in all five floating plant genera under a single set of nutrient temperature conditions, future research efforts should systematically examine floating plant response diversity in a larger number of clones and environmental conditions. Future work should also consider variability in environmental conditions and species composition through space and time. Floating plants are expected to be easily dispersed by waterfowl and other vectors (*Barrat-Segretain, 1996*); therefore, species composition in a waterbody may change through time, with possible consequences for floating plant dominance.

## CONCLUSIONS

This study has identified differences in three floating plant species common to the northeast United States. Although species differences existed in the laboratory, there was no statistical support that the species richness of floating plants increases their dominance in the field. Although free-floating plants can be viewed as both a nuisance and an opportunity for applied uses, understanding the species-specific responses of these plants to abiotic conditions is essential for both management and applications.

## ACKNOWLEDGEMENTS

I would like to thank Ishmael Rahim and Eunice Asare for help with the laboratory experiments, Greg Bugbee and the Connecticut Agricultural Experiment Station for data access, and the Frank Melville Memorial Park Foundation, numerous property owners, and the Suffolk County Parks Department for access to water bodies. This manuscript was improved by feedback from Dianna Padilla, Stephen Baines, Heather Lynch, and Gary Mittelbach.

### Funding

This work was funded by the Lawrence B. Slobodkin Graduate Research Fund. The funders had no role in study design, data collection and analysis, decision to publish, or preparation of the manuscript.

## Competing Interests

The authors declare that they have no competing interests.

## Author Contributions

- Michael J. McCann conceived and designed the experiments, performed the experiments, analyzed the data, contributed reagents/materials/analysis tools, wrote the paper, prepared figures and/or tables, reviewed drafts of the paper.

## Data Deposition

The raw data has been supplied as Supplemental Dataset Files

## Supplemental Information

Supplemental information for this article can be found online at http://dx.doi.org/10.7717/peerj.1781#supplemental-information.

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
