# Peer review of "Response diversity of free-floating plants to nutrient stoichiometry and temperature: growth and resting body formation"

_PeerJ, doi:10.7717/peerj.1781_

## Round 0.1 · original submission · Minor Revisions

I'm pleased to note that both reviewers recognized the merit of your manuscript, and I am confident it will be acceptable with only minor revision. Please address points recognized by the reviewers, and let me know if you have any questions.

·

Basic reporting

- The author would benefit from citing a study from 2015 on the relative in vitro growth rates of duckweed (Ziegler et al. 2015 Plant Biology).

Experimental design

- The author should explain why the replicates for Exp I and II different. Briefly explaining the reasoning is sufficient.

- Author needs to explain why plants were grown at 30 C in Experiment II, and not at the range of temperature as in Exp I. Although maximum growth rates was the stated reasoning, the author states in the Discussion that species had similar growth rates under high temperatures, and species differences were more likely to be recorded under lower temperatures. I was curious as to why a range of temperatures with various nutrient stoichiometries was not used in Exp II, and I expected a single experiment combining the nutrient combinations and temperatures, which would essentially be Exp I and II together. I was unclear as to why these studies were conducted in two separate experiments, and I think more justification is needed.

Validity of the findings

- I think the author needs to address in the Discussion how the results may impact applied uses of floating plants. This argument was highlighted in the Introduction, and should be reflected in the Discussion/Conclusion as well. Speculation as to which species could be advantageous would be welcome. I think this was the biggest missing piece in the Discussion/Conclusion sections.

Additional comments

- Overall, this manuscript was very well written and easy to read.

- Figures were easy to interpret, simplified, and relevant to the results.

- I appreciated the clear explanation of methods in the appendices.

- I have included an annotated PDF for briefer comments and edits.

·

Basic reporting

The article is well written and the reporting is clear. The background information is appropriate to orient the reader, and the biological significance of the system and the rationale for the study are explained to show how the work fits into the broader field of knowledge. Citations are included and are appropriate. The structure of the article fits the template, and tables and figures are appropriately expressed. This is a coherent body of work.

Experimental design

The article reports original primary research, and the stated objectives are relevant. The methods are clearly reported and reproducible. Statistical analyses seem appropriate although I am not aware of an appropriate use of systematic assignment of treatments (L110).

Validity of the findings

The data are robust and clearly reported. The conclusions are clearly stated and reflect the objectives of the study. Limitations of the study are clearly stated, and interpretation of the results are appropriate. The need for future studies are included and provide direction for the further elucidation of response diversity in floating plant species.

Additional comments

Title
L 1 uses the term "free-floating". However, in the body of the paper he primarily uses the term "floating". These terms should be reconciled.

Abstract
L 23 "floating plant polycultures were not more dominant": awkward wording

Introduction
L 77 The surveys were not used to determine response variability. They were used to address the second objective regarding plant dominance.

Material & Methods
L 106 Insert "fronds" after "white or brown".
L 108 refers to "many previous studies": Author should cite more than one.
L 119 Indicate which component of "growth"; i.e., surface area
L 152 refers to "many experiments": Author should cite more than one.

---

## Round 0.2 · accepted · Accept

Thanks for responding to the reviewers' suggestions and questions in your revision.

·

Basic reporting

My suggestions were adopted and I am satisfied with the quality of the paper.

Experimental design

The additional explanation strengthens the methodology section.

Validity of the findings

The data reported and conclusions made are sound.

Additional comments

The changes to the manuscript have improved the value to the reader.